# Technical pipeline for screening microbial communities as a function of substrate specificity through fluorescent labelling

Shaun Leivers [1✉], Leidy Lagos[2,3], Philipp Garbers [1], Sabina Leanti La Rosa [1,2] & Bjørge Westereng [1✉]

The study of specific glycan uptake and metabolism is an effective tool in aiding with the continued unravelling of the complexities in the human gut microbiome. To this aim fluorescent labelling of glycans may provide a powerful route towards this target. Here, we successfully used the fluorescent label 2-aminobenzamide (2-AB) to monitor and study microbial degradation of labelled glycans. Both single strain and co-cultured fermentations of microbes from the common human-gut derived Bacteroides genus, are able to grow when supplemented with 2-AB labelled glycans of different monosaccharide composition, degrees of acetylation and polymerization. Utilizing a multifaceted approach that combines chromatography, mass spectrometry, microscopy and flow cytometry techniques, it is possible to better understand the metabolism of labelled glycans in both supernatants and at a single cell level. We envisage this combination of complementary techniques will help further the understanding of substrate specificity and the role it plays within microbial communities.

[1] Faculty of Chemistry, Biotechnology and Food Science, Norwegian University of Life Sciences, Ch.M.Falsens vei 1, 1432 Aas, Norway. [2] Faculty of Biosciences, Norwegian University of Life Sciences, Oluf Thesens vei 6, 1433 Aas, Norway. [3] Skretting ARC, Sjøhagen 3, 4016 Stavanger, Norway. ✉email: shaun.allan.leivers@nmbu.no; bjorge.westereng@nmbu.no

The study of interactions between carbohydrates and bacteria has long been highlighted as crucial to understanding the genetic diversity and complexity of microbial communities present within the human gastrointestinal tract (GIT)[1]. In turn, this information helps us further our knowledge of how modifications within diet, through a subsequent change in glycan uptake and utilization, may impact on the gut microbiome[2]. Previous studies have emphasized the importance of a symbiotic environment[3] and discussed competition and cooperation[4] behaviours in the human gut microbiome. Whilst a number of known beneficial members of the gut, particularly those belonging to *Lactobacillus* and *Bifidobacterium* spp., have been routinely studied with regards to uptake and metabolism of simpler glycans in recent years[5,6], those assigned to the genus *Bacteroides*, which accounts for ~25% of bacterial cells in the human intestine[7], have only more recently garnered such attention for their capability to utilize complex glycans[8,9]. The possibility to develop complex prebiotic glycans to selectively engage beneficial microbes rests on understanding substrate selectivity as bacteria have developed highly specialized mechanisms for catabolizing the vast ensemble of glycan structures and their associated motifs[10,11].

Members of the Bacteroidetes phylum are renowned for their abilities to degrade a wide range of glycans available through diet, host secretions, microbial exopolysaccharides, and capsules[9]. Within Bacteroidetes, *Bacteroides* dedicate a substantial proportion of their genome to carbohydrate active enzymes (CAZymes)[12]. Sequencing of the *B. thetaiotaomicron* (hereafter referred to as *B. theta*) genome[13] along with the discovery of the architecture of polysaccharide utilization loci (PULs)[14], by which the Bacteroidetes depolymerize and degrade complex glycans, opened the door for further understanding of the degradation mechanisms[15] of glycans in the gut[16]. The first PUL that was identified and fully characterized was the *B. theta* Starch utilization system (Sus), which consists of eight genes (*susRABCDEFG*) coding for proteins involved in the sensing, capture, uptake, and hydrolysis of starch[17]. The Sus has been used as an archetype for understanding other PUL systems and the nomenclature used to refer to the Sus is the convention when describing other loci. Despite variations in the glycan they target, PULs are defined by the presence of so-called SusC-like and SusD-like genes, encoding an outer membrane TonB-dependent transporter and a binding protein, respectively. SusCD-like complexes have been shown to mediate substrate uptake via a "pedal bin" transport mechanism, where SusC forms the barrel of the bin and the SusD-like protein acts as a lid on top of the barrel, opening and closing to facilitate substrate binding[18]. A few studies have reported that SusCD transporters have a size limit for substrate transport, indicating that ~5 kDa may be a general total size limit for these systems[18–20]. More recently, several PULs systems from other phyla such as Firmicutes and Bifidobacteria have been studied in detail[21–23]. Whilst a wealth of knowledge is available regarding the individual strain-based degradation processes involved in complex glycan catabolism[24,25], the effect of competitive environments faced by bacteria, despite recent advances in the field[26,27], is poorly understood. With this in mind, a set of fast and effective tools for screening and testing different substrates to identify which bacteria within a community are capable of catabolizing them is of great interest.

Interspecies competition between *Bacteroides* spp. has been studied in mice fed "microbiota directed foods" (or MDF, through bead-based labelling)[27]. Such data has thus far enhanced the understanding and potential for therapeutic targeting of beneficial human gut bacteria through the use of prebiotic introductions[22]. Labelling of *B. fragilis* using 'metabolic oligosaccharide engineering (MOE) and biorthogonal click chemistry (BCC)'—used in vivo to observe host–commensal interactions in the intestine were able to further label and resolve mixed bacterial species including *B. vulgatus*, *B. ovatus*, and *B. theta*[28]. The deletion of specific PULs for degradation and transport of certain glycans[29], as well as identifying the storage of labels within certain cells[30], was investigated through fluorescent labelling of cells, specifically utilizing epifluorescence microscopy and super-resolution structured illumination microscopy (SR-SIM). *B. cellulosilyticus* WH2 along with *B. caccae*, *B. ovatus*, and *B. theta* have been used to explore symbiotic relationships in the gut, focused mainly around xylan utilization[31]. Discrete chemical structure variation can be employed as a directive influence on gut microbiome short chain fatty acid (SCFA) production whilst microbiome-modulating strategies based on variation in fine structure of carbohydrates were found to confer a selective enrichment of a specific number of bacterial taxa[32]. The application of dietary carbohydrates in relation to food composition, gut microbiota and metabolic outputs as well as the interaction and utilization of functional groups is key to unravelling this complex community[33–35].

Fluorescent labelling of substrates is a valuable tool in furthering the understanding of the glycobiome[36]. The importance and impact of substrate decoration and the relationship to selectivity has been explored through systematic depolymerization and labelling of carbohydrates, such as β-glucans[37], xylan[38] cellulose[39], and complex *N*-glycans (CNGs)[25]. Applications of fluorescently labelled substrates have allowed for the study into the uptake and utilization of polysaccharides in marine microbial communities[30,40,41]. However, the overwhelming majority of these studies employ a bulky and expensive fluorescein-based derivative such as fluorescein isothiocyanate (FITC)[36], fluorescein-5-thiosemicarbazide (FTSC)[42] or fluorescein amidite (FAM) as the fluorescent component. Despite extensive reviews into the capabilities of a multitude of readily available fluorescent glycoconjugates[43–46], the experimental pool of studies which use a non-fluorescein based alternative label in microbiome analysis is still limited, with the exception of 2-($N$-(7-Nitrobenz-2-oxa-1,3-diazol-4-yl)amino)−2-deoxyglucose (2-NBDGlucose) which is routinely used for cell tracking in yeast[47,48] and *Escherichia coli*[49], and more recently in the analysis of the uncultured rumen microbiome[50].

Despite the evident potential for implementing glycan labelling as a tool for microbial exploration, the focus has mainly, in terms of identification and quantification[51] been on enhancing chromatographic techniques, specifically high performance liquid chromatography (HPLC)-ultra violet (UV)[52] and HPLC-fluorescence detection (FLD)[53]. In relation to efficient separation, hydrophilic interaction chromatography (HILIC) columns have proven useful for analyzing fluorescently labelled glycans, providing high selectivity for complex oligosaccharides[54,55]. When analyzing labelled carbohydrates mass spectrometry (MS) or comparable hyphenated techniques (LCMS) have provided an effective platform for the elucidation of complex, highly structurally similar glycans[56–58]. Flow cytometry has emerged as a powerful tool to disseminate labelled cells by fluorescence-activated cell sorting (FACS)[50]. Fluorescently labelled glycans used in microbial fermentation experiments have been observed through a variety of microscopy techniques, particularly epifluorescence in combination with more detailed studies by confocal microscopy[29,30].

Use of small aromatic fluorophores has commonly and successfully been applied in glycosylation studies[59,60], to enable effective analytical glycan detection. Two such readily used are 2-aminobenzoic acid (2-AA) and 2-aminobenzamide (2-AB). These labels are added to glycans via reductive amination[61], although these methods have often included a number of drawbacks such as the use of cyanide (CN) containing reducing agents

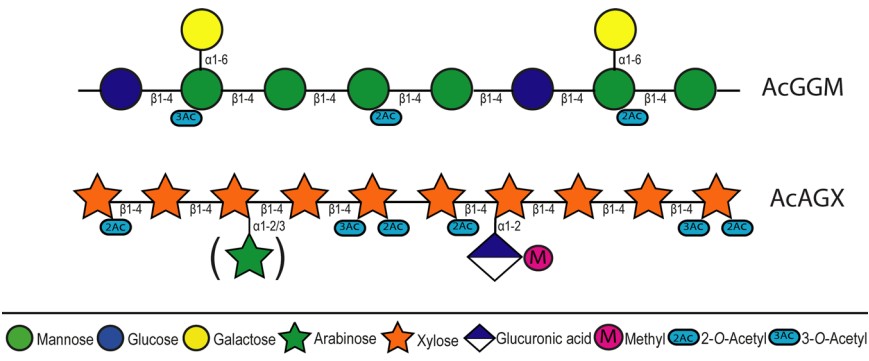

**Fig. 1 Schematic of substrates used in this study.** Acetylated galactoglucomannan (AcGGM) from Norwegian spruce wood and acetylated (arabino) glucuronoxylan (AcAGX) from Norwegian birch wood (very small amount of arabinose present).

as well as using dimethyl sulfoxide (DMSO) as the reaction medium. Due to its high toxicity, eliminating residual cyanide in products is essential when utilizing labelled glycans in microbial fermentations. More recently, alternative methodologies using 2-Picoline-borane (pic-BH₃) in one pot reductive aminations in methanol, water and solvent free conditions have been explored[62]. Comparisons and potential benefits of pic-BH₃-based reactions over more traditional NaCNBH₃ have also been extensively demonstrated[63–65]. Many glycans have non-carbohydrate motifs, such as acetylations, phosphate-, sulfate- or methyl- groups that are potentially vulnerable to pH extremes. Choosing benign labelling conditions which allow such substituents to remain intact after conjugation is thus of key importance. Smaller fluorescent compounds like 2-AB may affect binding and transport to a lesser extent than more bulky fluorophores, thus potentially providing a sensible complementary alternative to fluorescein-based derivatives for labelling.

Here we look to introduce a simple, efficient and cheap alternative use of the fluorescent label 2-AB to be utilized as a tool for monitoring glycan uptake in relation to bacterial substrate specificity. The procedure is both substituent benign and utilizes non-toxic reagents that ensure substrates are readily accessible in a biological context. We also present a comprehensive analytical approach that warrants highly sensitive, routine HPLC principles combined with FACS and high-resolution microscopy.

## Results and discussion

Many glycan-labelling techniques used for biological visualization purposes focus on large, bulky multi-ring moieties. In contrast, fluorescent labels used to compensate for a carbohydrates lack of natural chromophore in analytical, specifically chromatographic applications, are often smaller and subsequently less complex. Particularly important for practical applications, these labels are often cheaper and can be more readily analytically identified than labels/dyes produced specifically for biological applications. However, whilst structurally larger fluorescent labels such as fluorescein or pyrene/anthracene-derived compounds have also been used for chromatographic applications, conversely smaller labels, such as 2-AB, have not been routinely utilized when studying biological environments.

### Substrate generation and characterization. In this study, we generate fluorescently labelled carbohydrates using β-mannan-(AcGGM) and xylan-derived (AcAGX) oligosaccharides from Norway spruce and birch wood, respectively (Fig. 1), in reducing end coupling by reductive amination (via picoline borane) to successfully conjugate 2-AB molecules (Fig. 2a). Modified versions of each of these substrates were created by treatment with a β-mannanase (R. intestinalis β-mannanase RiGH26) for AcGGM

generating the shorter GH26-AcGGM and a combined alkaline and xylanase treatment for AcAGX, generating the shorter, deacetylated GH10-AGX. The labelled products were subsequently detected and differentiated by HPLC-HILIC-FLD (Fig. 2b), with MS and tandem MS (MS/MS) (Fig. 2c) and matrix-assisted laser desorption ionization time-of-flight (MALDI-ToF) MS (Fig. 2d, e). Initially, this methodology was optimized using commercially available manno- and xylooligosaccharides (M/X 1–6). The products generated at this initial stage were used as standards going forward (Supplementary Fig 1). The study also extended to a number of other monomers (glucose), disaccharides (lactose), and oligosaccharides (enzymatically and chemically modified mannans and xylans), observing that this approach is applicable to a wide range of glycans with reducing ends. Retaining structural motifs and decorations, such as, in this case acetylations, was seen as an essential attribute for the methodology. In the characterization procedure we observed no appreciable losses of acetylations in the labelled substrates produced as observed in Fig. 2d.

### Substrate purification and excess label removal. The purification process is of particular importance in this methodology. Numerous studies have reported on the post-labelling purification of 2-AB glycoconjugates, with varying degrees of both success and expense. After trialling numerous solvent and solid phase extraction (SPE) systems (Supplementary Note 1), our method of choice for substrate purification and excess label removal was liquid–liquid extraction with ethyl acetate. This approach provided clean, almost excess label-free extracts which could be efficiently recovered by drying in vacuo. Ethyl acetate was subsequently employed for the extraction of free label in solution, whilst not as efficient as octanal it provided an effective reduction of excess label (Supplementary Fig. 2).

These experiments were initially carried out on milligram scale and subsequently scaled up (Supplementary Fig. 3) to produce multi-gram-labelled material, therefore enabling this experimental methodology to be efficiently scaled up for bioreactor applications and potential in vivo studies.

### Fermentations and growth. To understand both bacterial growth conditions and the optimum conditions for label uptake, the inclusion of labelled substrates to bacterial cultures was performed in one of four ways (Fig. 3a). Condition set 2, consisting of a mixture of glucose (on which all bacteria grew effectively) and labelled substrate, was chosen going forwards as it provided the best resolution for flow cytometry measurements, as poor or in some instances, no growth on labelled and/or unlabelled substrates (conditions 1, 3 and 4 (for B. caccae and B. theta)) resulted in a lack of viable cells to analyse.

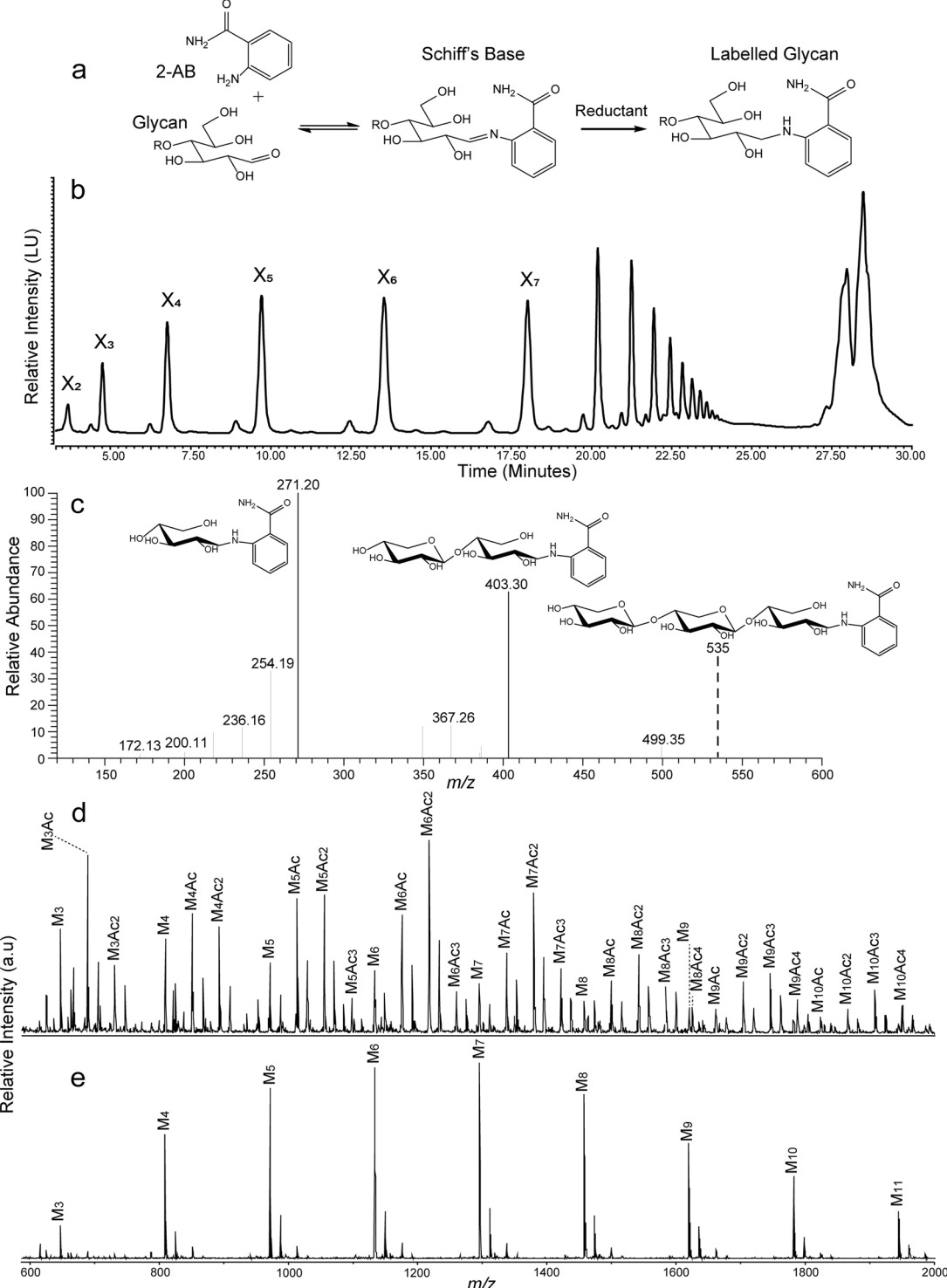

**Fig. 2 Synthesis and characterization of 2-AB-labelled glycans. a** Schematic of synthetic pathway for 2-AB labelling of glycans. **b** HPLC-HILIC-FLD chromatogram of 2-AB labelled xylan-derived oligosaccharides. **c** Corresponding MS and MS/MS fragmentation of 2-AB-labelled xylotriose—major fragments are 2-AB xylobiose and 2-AB xylose. **d** MALDI-ToF identification of 2-AB labelled AcGGM—the spectra clearly demonstrates the retention of acetylations after both initial labelling and purification. **e** MALDI-ToF identification of 2-AB AcGGM (post-labelling deacetylation). In panels **b**, **d**, **e** the following abbreviations are used: X xylose unit, M mannose unit, Ac acetyl group.

Figure 3b shows an example of a comparative set of growth curves for *B. cellulosilyticus* grown on labelled and unlabelled AcGGM, as well as glucose (control) and 2-AB (negative control). While there is no growth solely on the label, there is comparative growth on the labelled and unlabelled substrate.

**T0 sampling**. Metabolism of labelled carbohydrates in bacterial fermentations was followed by incremental analysis of the isolated supernatant by HPLC-HILIC-FLD. The bacteria used were selected based on their known ability to utilize β-mannans and xylans (*B. cellulosilyticus* and *B. ovatus*[2]) or the lack of this capability (*B. theta* and *B. caccae*[2,66]). Supernatant product profiles

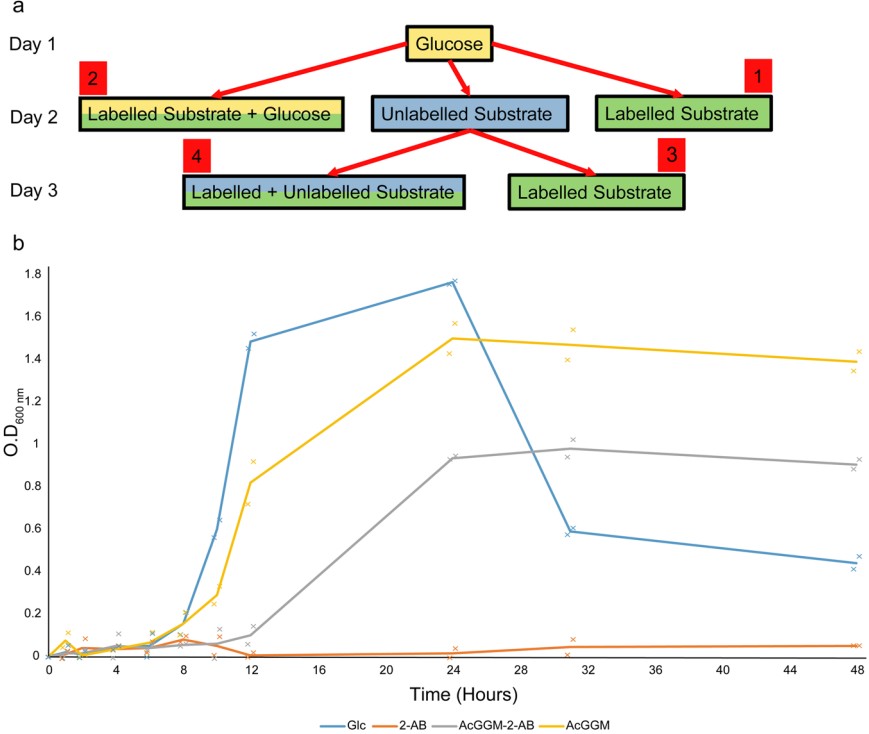

**Fig. 3 Plan implemented to determine the most efficient method of identifying labelled glycan uptake and bacterial growth curves for *B. cellulosilyticus*. a** Condition set **1** added to overnight glucose-grown culture, **2** introduced as a mixture of glucose and labelled substrate to overnight glucose-grown culture, **3** overnight glucose-grown culture supplemented with unlabelled substrate, followed by next day addition of either labelled substrate or **4** a mix of labelled and unlabelled substrates. Glucose (yellow), unlabelled (blue), labelled (green). **b** *B. cellulosilyticus* grown on glucose (blue), 2-AB (orange), 2-AB-labelled AcGGM (grey), and AcGGM (yellow) (data based on 2 biological replicates).

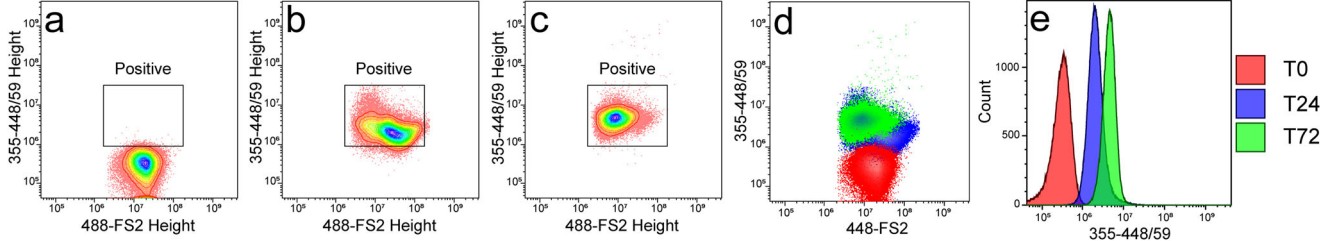

**Fig. 4 Flow cytometry of *B. cellulosilyticus* grown on AcGGM. a–d** Gating strategy developed based on 355 nm Height (2-AB) vs. 488-FSC2 Height (DNA-dye), clearly observed shifts based in fluorescence of bacteria grown on unlabelled glycans **a** T0, compared to bacteria grown on labelled substrates for T 24 **b** or T 72 h **c. d** The overlapped dot-plot demonstrates the incorporation over time by increasing levels of fluorescence (labelled cells. *B. cellulosilyticus* grown on AcGGM) at 24 h (blue) and 72 h (green) compared to the unlabelled substrate (red). **e** Histogram indicating the shift in fluorescence observed for cells grown on labelled substrates for 24 h (blue) and 72 h (green) compared to the unlabelled substrate (red).

were compared to both standards and negative control samples. Negative controls assigned as T0 samples, consisted of substrates dissolved in fermentation media. True'T0' samples, i.e. samples acquired directly after inoculation of fermentations with labelled substrate, were deemed unfeasible to be used for comparative studies; this was due to rapid labelled carbohydrate uptake and catabolism, even within the relatively short time period of exposure during sampling.

**Flow cytometry**. Evaluation of uptake and incorporation of labelled components by bacterial cells isolated from fermentation experiments by flow cytometry led to a clear and distinct identification of labelled cells. A marked shift in fluorescence demonstrated by the implemented gating strategy (355 nm height vs. 488-FSC2 height), conclusively confirmed the presence of 2-AB (Fig. 4a–d). The gating strategy incorporated a DNA dye (see the section "Confocal methodology" in the "Methods"

section) to ensure only cells were included in the overall identification. Visualization of the shift in fluorescence and therefore increase in labelled glycan uptake could be more easily displayed and therefore tracked using the plot of 355 nm outlined in Fig. 4e. It should be noted that in numerous experiments rapid uptake was readily observed, generally though the method could be used effectively to monitor the uptake of labelled glycans in bacterial cells over time.

**HPLC-HILIC-FLD**. Uptake and utilization of labelled substrates could be effectively monitored over time by analysis of supernatant sub-samples taken from ongoing anaerobic fermentations (Fig. 5). Supernatant samples were compared directly with the labelled starting material added in the initial stages of the fermentation. The substrate used for comparison was also subjected to the same conditions applied in fermentations, with the almost identical chromatogram (compared to T0 chromatogram)

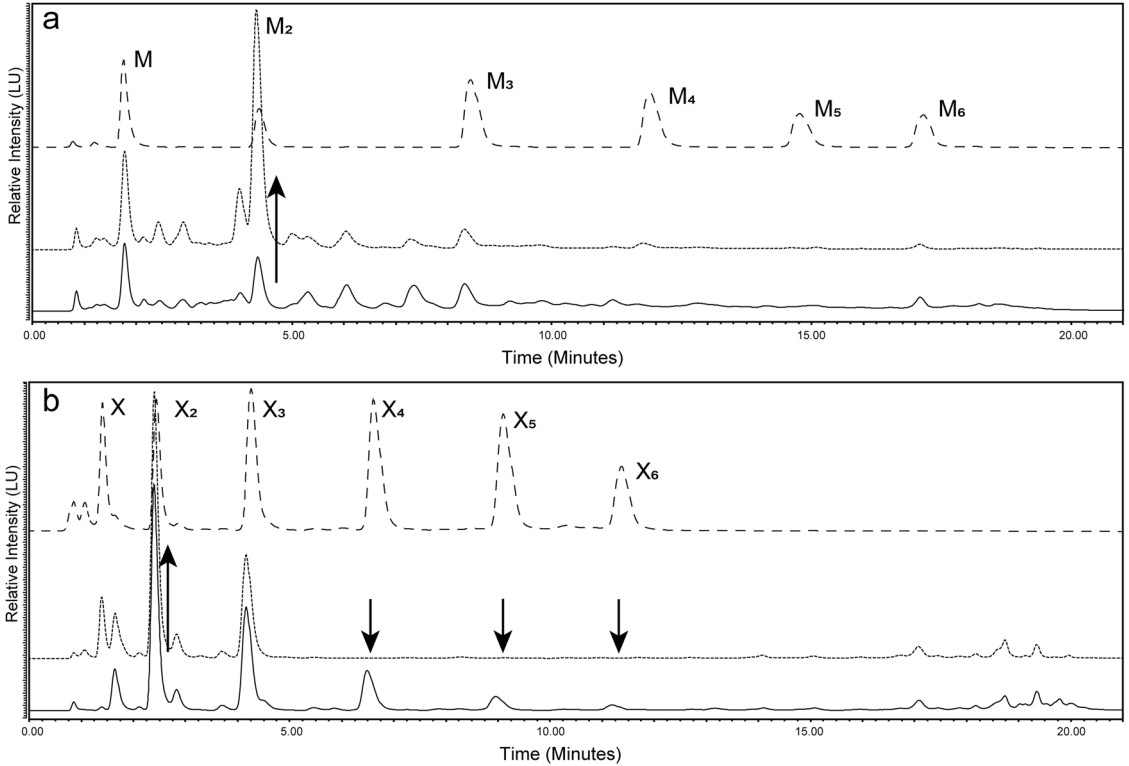

**Fig. 5 Monitoring of fluorescently labelled substrate metabolism by HPLC-HILIC-FLD. a** *B. cellulosilyticus* grown on GH26-AcGGM—Mannose 1–6 standards (dashed line), 2-AB-labelled GH26-AcGGM (dotted line) and 24 h supernatant sample (solid line). A clear increase (upwards arrow) in mannobiose is observed whilst an overall reduction in complexity is also noticeable. M mannose, $M_2$ mannobiose, $M_3$ mannotriose, $M_4$ mannotetraose, $M_5$ mannopentaose, $M_6$ mannohexaose. **b** *B. cellulosilyticus* grown on 2-AB-labelled GH10-AGX—Xylose 1–6 standards (dashed line), 2-AB-labelled GH10-AGX (dotted line) and 24 h supernatant sample (solid line). A clear reduction in levels of xylotetraose, xylopentaose, and xylohexaose (downwards arrows) was observed whilst an accumulation of xylobiose (upwards arrow) was highly pronounced. X xylose, $X_2$ xylobiose, $X_3$ xylotriose, $X_4$ xylotetraose, $X_5$ xylopentaose, $X_6$ xylohexaose.

observed thus confirming that any observable degradation occurred as a direct result of bacterial processing of the carbohydrates and not due to natural degradation over time from exposure to the controlled anaerobic environment.

A combination of synthetically prepared standards (Supplementary Fig. 1) and, when required, MALDI-ToF, were applied to characterize the degradation products present in supernatant chromatograms. In either single strain or co-cultured fermentations, the degradation (or lack thereof) of substrates, observed by the shift of oligosaccharide degree of polymerization (DP) from high to low, was clearly demonstrated through chromatographic and mass spectrometry techniques (single strain—Figs. 5, 6c and Supplementary Figs. 4a, b and 5b, co-cultured—Supplementary Fig. 5c).

The overall technique developed in this study is demonstrated in Fig. 6, using a single strain (*B. cellulosilyticus*) and a single 2-AB-labelled carbohydrate substrate, GH10-AGX. The bacterial strain grown on the labelled substrate could be effectively monitored over time. Flow cytometry (Fig. 6a, b) was able to display, through an increase in fluorescent response over time, the higher incorporation levels of the labelled substrate within the cells. While chromatographically, through HPLC-HILIC-FLD, monitoring could be employed to follow the contents of the supernatant and the increase or decrease in abundance of certain types and chain lengths of labelled carbohydrates over time (Fig. 6c). This may be utilized in a number of ways, e.g. to obtain high resolution information on microbial substrate preferences. Furthermore, the fluorescent labelling allows for the use of microscopy to obtain further insight into uptake.

**Confocal microscopy**. Isolated and fixed cells recovered from 2-AB-labelled fermentations could be positively screened for and further characterized by confocal microscopy (Fig. 7). DAPI (4′,6-diamidino-2-phenylindole) is commonly used for the staining of DNA in microscopy techniques. However, 2-AB—Ex 330 Em 420, has an overlapping Ex/Em range—DAPI Ex 358 Em 461. Two alternative dyes were therefore investigated for DNA staining including SYTO 9 and SYBR Green I[67,68], along with two potential stains for membrane identification (FM 5-95[69] and *Bac*Light Red[70]). SYBR and SYTO 9 both performed efficiently in the staining of bacterial DNA and were used interchangeably for both confocal microscopy and flow cytometry. For cell wall/membrane staining *Bac*Light Red provided marginally more efficient coverage and therefore improved resolution when compared to FM 5–95 in microscopy applications.

**Epifluorescence microscopy**. Alongside confocal microscopy labelled bacterial cells were additionally identified by standard epifluorescence microscopy. The dedicated wavelength for DAPI excitation was used in order to stimulate and view the internalized 2-AB label. Bacterial cells cultured on either mannan or xylan-based substrates were clearly identifiable in samples after only 1 h (Fig. 8). In order to visualize non-labelled cells, the bright field application was utilized to demonstrate the lack of fluorescence in cells grown without 2-AB-labelled substrates. Identification of labelled bacteria through epifluorescence could be routinely employed as a comparatively

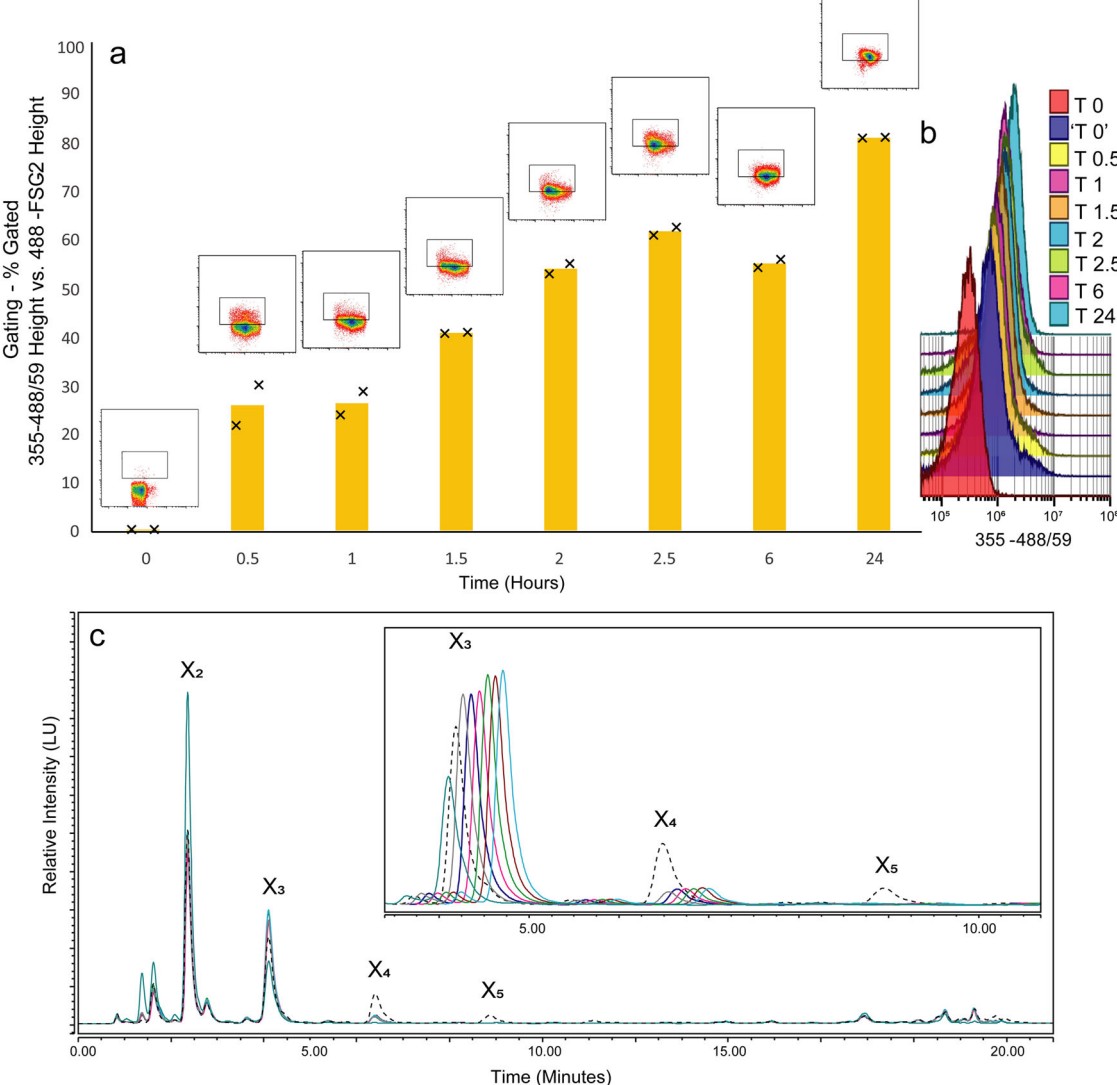

**Fig. 6 Monitoring of labelled glycan uptake and degradation over time using a combination of flow cytometry (cell tracking) and HPLC analysis (supernatant tracking)—B. cellulosilyticus grown on 2-AB-labelled GH10-AGX. a** The graph shows an increase in the number of fluorescently labelled cells from flow cytometry measurements (data based on two biological replicates), viewed as a percentage of double positive cells (355/488 nm), through a range of sampling time points, up to 24 h. **b** Shift in fluorescence (2-AB 355 nm) at all recorded time points. 'T0' is included as to demonstrate the initial, almost immediate shift that arises from addition of the labelled substrate. **c** HPLC-HILIC-FLD chromatograms demonstrating the degradation and consumption of the labelled substrate (dotted line) over time on the supernatant of the corresponding cells analysed by flow cytometry—chromatograms are shifted to more clearly show the difference in peak heights. $X_2$ xylobiose, $X_3$ xylotriose, $X_4$ xylotetraose, $X_5$ xylopentaose.

quick and efficient methodology for labelled-substrate growth confirmation.

**A technique for screening microbial communities/substrate specificity**. The labelling methodology was further explored through the use of complex glycans. *B. ovatus* demonstrated incorporation of labelled substrates to differing degrees (after 24 and 72 h) (Fig. 9). More complex glycans, like AcGGM and AcAGX, both observed the highest uptake levels whilst GH10-AGX and GH26-AcGGM showed much lower levels of incorporation (Fig. 9b, d). Expanding the experiment focus out to include a range of *Bacteroides* strains, co-cultured with *B. ovatus*, this trend was also widely observed (Fig. 9). However, while chromatographic and MS data generally corresponded well with the accompanying label incorporation trends identified by the flow numbers, in some cases, despite low inclusion levels

observed in the flow, greater levels of glycan degradation was observed when studied by MALDI-ToF and HPLC-HILIC-FLD (Supplementary Fig. 4a, b).

The effect of substrate complexity on individual bacteria and co-cultures was further analysed by using a single substrate, AcAGX (Supplementary Fig. 5).

Further advancement of labelling capabilities may be achieved by employing both chemical and enzymatic techniques to enhance and expand glycoconjugates for potential microbial consortia analysis[71,72]. Recent studies have indicated that increased consideration is required when selecting substrates to study microbial interactions[73,74]. Studying substrate specificity through a labelling-based approach could help bring a heightened level of knowledge to this area of research. Including other types of labels, specifically a fluorescein-based labels such as FITC as well as 2-AB could help to shed light on the way in which glycans are internalized and degraded as part of a bacterial

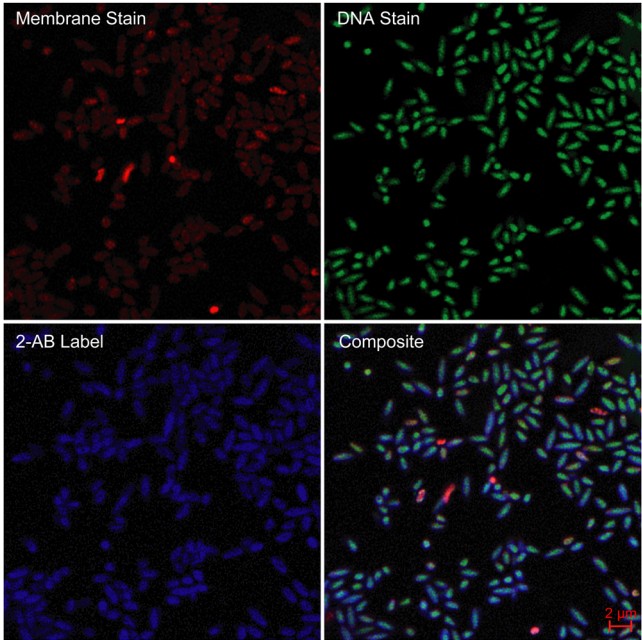

**Fig. 7 Confocal microscopy of 2-AB-labelled cells recovered from *B. cellulosilyticus* grown on GH26-AcGGM after 1 h.** *Top right* DNA view of cells stained with SYBR Green I. *Top left* Membrane view of cells stained with *Bac*Light Red. *Bottom left* View of cells labelled with 2-AB. *Bottom right* Composite view of all three channels.

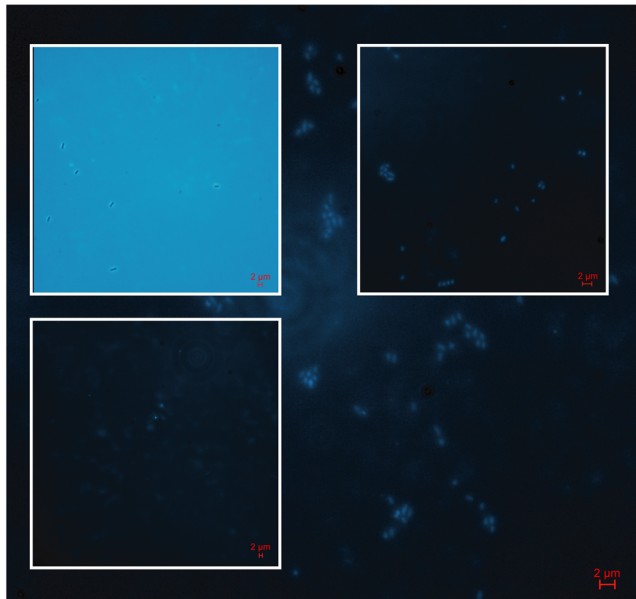

**Fig. 8 Epifluorescence microscopy of 2-AB-labelled cells recovered from *B. cellulosilyticus* grown on GH26-AcGGM and AcAGX.** *Main body*—Clusters of 2-AB-labelled cells grown on GH26-AcGGM (positive). *Inset bottom left*—View of equivalent unlabelled cells (negative). *Inset top left*—Brightfield view of equivalent unlabelled cells (negative). *Inset top right*—Clusters of 2-AB-labelled cells grown on AcAGX (positive).

consortia's catabolism process. We have not experienced any impact on uptake, imparted by the addition of the 2-AB label, with most growth curves for bacteria with labelled and unlabelled substrate showing relatively comparable optical densities (ODs), Supplementary Fig. 6. Having a wider variety of glycan label alternatives (large, small, flexible, charged) may also help to

understand and/or reduce any possible bias that the incorporation of a label may induce. Potential bias could also be addressed by the systematic addition of labels in different positions (on glycans). The addition of labels at the reducing and non-reducing end would further the currently limited studies into the potential directionality[72] of glycan uptake by microbiota. Direct molecular imaging of glycans could be seen as a logical next step and complementary technique to labelling-based approaches[75].

High throughput screening of large numbers of substrates with vast numbers of different microorganisms could be achieved through an integrated plate-based process, linked to a dedicated database system[76], ultimately towards an automated glycomic platform[77]. Metagenomics and metaproteomics approaches, in combination with growth and biochemical analysis to assess complex polysaccharide degradation[41], is seen as a key next step for furthering this technique. Additionally, heading towards an automated process for identifying specific subgroups of microbial groupings within complex communities, a process such as the one outlined here could be integrated into a fully rounded 'omics approach[78] with relative ease.

Incorporating cell sorting, coupled with qPCR analysis to enable the continued exploration into understanding glycan metabolism on a genomic level will be further explored. It is envisaged that this technique as a whole could be utilized for selective sorting and identification of microbial communities in relation to microbes' glycan selectivity.

Our combination of techniques and compatible methodologies, implementing a small fluorescent label such as 2-AB (approximately the size of a monosaccharide) for the monitoring and study of glycan uptake allows for the continued development into the high-resolution analysis of microbial systems. In demonstrating this scalable, non-toxic, process of fluorescent coupling, applicable to several types of mono-, oligo- and polysaccharides which include prominent structural features, such as acetylations, we have provided a framework for the screening of naturally existing substrates by several complementary techniques. The addition of a fluorophore should help to reduce the number of targets when analyzing large microbial consortia, allowing for a simpler analysis of microbiota compositions, therefore leading to a higher level of precision than is commonly achieved with conventional methodologies.

## Methods

**Substrates**. Mannobiose, mannotriose, mannotetraose, mannopentaose and mannohexaose, xylobiose, xylotriose, xylotetraose, xylopentaose and xylohexaose were from Megazyme (Ireland). Mannose, xylose, 2-picoline borane, 2-aminobenzamide, ethyl acetate, methanol and ammonium formate were purchased from Sigma-Aldrich (Germany).

Acetylated galactoglucomannan (AcGGM) from Norway spruce (*Picea abies*) was produced in house from dried wood chips[79]. A simplified (lower DP range) version of this substrate, named GH26-AcGGM was produced by treating the AcGGM with a β-mannanase (*R. intestinalis* β-mannanase *Ri*GH26).

Acetylated (arabino)glucuronoxylan (AcAGX) was produced in house from birch (*Betula pubescens*) chips[80]. A simplified (lower DP range and deacetylated) version of this substrate, named GH10-AGX was produced by treating the AcAGX with sodium hydroxide to remove all acetylations followed by subsequent treatment with the commercial xylanase Shearzyme (Novozymes, Denmark).

**Procedure for 2-AB labelling of mono- and oligosaccharides as standards**. Reductive amination-based labelling of both mono- and oligosaccharides with 2-AB was loosely based on the original methodology devised by Bigge et al. 1995[61]. However, in this method DMSO was replaced by aqueous acidified methanol and NaBH$_3$CN with Pic-BH$_3$ as reported by Vanina et al. 2011[64].

**Small scale samples for standards**. In an Eppendorf tube mannotriose (0.5 mg, 1 umol, 1 equivalent) was dissolved in an amount of H$_2$O to which a freshly prepared aliquot of Pic-BH$_3$ (0.52 mg, 5 µmol, 5 eq) in methanol was added. To this solution an amount of 2-AB (0.16 mg, 1.1 µmol, 1.1 eq) in methanol was added along with a volume of acetic acid to achieve a final solution ratio of 35:50:15 methanol, water, acetic acid, respectively (v/v/v). The tube was heated at 60 °C for 2 h and constantly

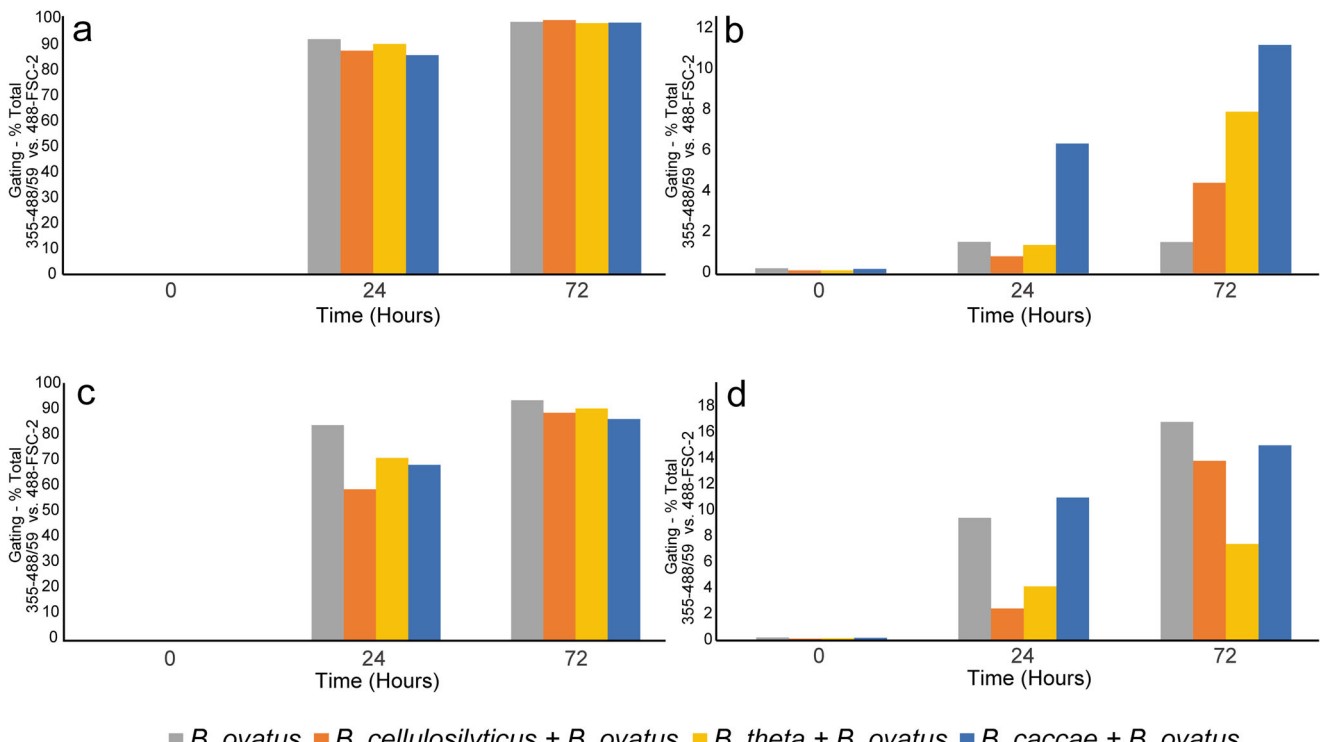

**Fig. 9 Screening of labelled glycan uptake as a function of substrate specificity in co-cultured bacterial fermentations measured by flow cytometry. a** AcGGM—considerable uptake observed after 24 and 72 h for all strain combinations. **b** GH26-AcGGM—comparatively far lower uptake observed after 24 and 72 h for all strain combinations. **c** AcAGX—considerable uptake observed after 24 and 72 h for all strain combinations. **d** GH10-AGX—comparatively far lower uptake observed after 24 and 72 h for all strain combinations.

shaken at 500 rpm. After 2 h the solution was cooled and evaporated to dryness via CentriVap (Labconco, USA). Samples were then reconstituted in water and extracted three times with ethyl acetate to remove excess labelling reagent. Samples were then freeze-dried and stored as solids, with tubes wrapped in foil to prevent light degradation.

**Gram scale preparation of samples for fermentation usage**. In a 50 mL Falcon tube Mannan/Xylan-based polysaccharides (1 g, 1 eq) were dissolved in an amount of $H_2O$ to which a freshly prepared aliquot of Pic-$BH_3$ (0.42 g, 5 eq) in methanol was added. To this solution an amount of 2-AB (0.55 g, 5 eq) in methanol was added along with a volume of acetic acid to achieve a final solution ratio of 35:50:15 methanol, water, acetic acid, respectively (v/v/v). The tube was heated at 60 °C for 2 h in a water bath and constantly agitated. After 2 h the solution was cooled and evaporated to dryness via CentriVap (Labconco, USA). Samples were then reconstituted in water and extracted three times with ethyl acetate to remove excess labelling reagent. Samples were then freeze-dried and stored as solids, with tubes wrapped in foil to prevent light degradation.

**Multi-gram scale preparation of samples for fermentation usage**. In a 2 L round bottom flask Mannan/Xylan-based polysaccharides (12 g, 1 eq) were dissolved in an amount of $H_2O$ to which a freshly prepared aliquot of Pic-$BH_3$ (5 g, 5 eq) in methanol was added. To this solution an amount of 2-AB (6.6 g, 5eq) in methanol was added along with a volume of acetic acid to achieve a final solution ratio of 35:50:15 methanol, water, acetic acid, respectively (v/v/v). The flask was heated in an oil bath at 60 °C with constant stirring. After 2 h the solution was cooled, and methanol removed by rotary evaporation. Samples were then extracted three times with ethyl acetate to remove excess labelling reagent. Residual ethyl acetate was removed by rotary evaporation. The solution was transferred to 50 mL Falcon tubes and freeze-dried. The resulting solids were wrapped in foil to prevent light degradation.

**Confocal methodology**. Fluorescently labelled cells were prepared for visualization by adding 1 µL of the fixed cell solution and diluting with 999 µL PBS, then subsequently stained with 1 µL SYTO 9 (for DNA) and 1 µL BacLight Red (for cell wall) bacterial stain Ex581/Em644 (Invitrogen, Thermo Fisher, UK). For mounting, 2 µL of the resulting solution was then added to a poly-D-lysine-coated slide and dried. Slides were then washed with MQ water to remove excess salts and dried again. One drop of Fluoroshield mounting medium (Sigma Aldrich, UK) was then added to the cells followed by the addition of a cover slip. The cells were visualized

on a Zeiss LSM 800 confocal laser scanning microscope (Carl Zeiss, Germany) using the following nm lasers—561 (558–575) for cell wall, 488 (483–500) for DNA, and 405 (353–465) for 2-AB. All images were processed using the Zen Blue Edition software v.3.0 (Carl Zeiss, Germany).

**Epifluorescence methodology**. In addition to confocal microscopy, cells were also viewed and identified as fluorescently labelled by epifluorescence microscopy. Labelled bacterial cells (0.4 µL) were mounted onto agarose-coated (1.2%) glass slides and secured with coverslips. Unlabelled cells were also treated in the same way. Visualization was achieved using the DAPI wavelength (1.5 s exposure time) whilst negative control samples were also viewed using the brightfield to add further confirmation. Cells were observed on a Zeiss microscope (Carl Zeiss, Germany)—Axio Observer Z1/LSM700—HXP-120 Illuminator (fluorescence light source). Images were acquired using an ORCA-Flash4.0 V2 Digital CMOS camera (Hamamatsu Photonics) through a ×100 phase-contrast objective. All images were analysed with the Zen Blue Edition software v.3.0 (Carl Zeiss, Germany).

**Fermentations**. Human-gut-derived *Bacteroides* sp. (*B. thetaiotamicron* 7330[81], *B. ovatus* ATCC 8483[81], *B. cellulosyliticus* DSM 14838[82], and *B. caccae* ATCC 43185[83]) were cultured into freshly prepared minimal medium (MM)[84] supplemented with 5 mg/mL glucose. All fermentations were performed at 37 °C in an anaerobic cabinet (Whitley A95 Workstation; Don Whitley, UK) under an 85% $N_2$, 5% $CO_2$, and 10% $H_2$ atmosphere. Where fermentations were supplemented with labelled substrates, an aliquot of an initial overnight fermentation (as previously described) was taken and inoculated into a solution containing freshly made 2× MM (50% v) along with an amount of labelled substrate and equivalent glucose solution (50% v). Co-cultured samples were initially grown independently as described above; overnight samples were then combined into freshly prepared MM with 5 mg/mL glucose and grown overnight before the introduction of labelled substrates as described above.

**Fermentation sampling**. Samples taken from fermentations were processed immediately to minimize any changes to cell dynamics/conditions. 100 µL samples were recovered at each specified time point and transferred to Eppendorf tubes. Samples were then centrifuged (14,000 rpm, 4 min), after which the supernatant was frozen and stored at −20 °C for further analysis. The cells were then fixed by resuspending the cell pellet in 100 µL 2% paraformaldehyde (PFA)—4% solution diluted in PBS (Invitrogen, Thermo Fisher, UK) for 1 h at room temperature. After 1 h samples were centrifuged (14,000 rpm, 4 min) and the supernatant discarded. The pellet was then washed with

phosphate-buffered saline (PBS; pH 7.4—137 mM NaCl, 2.7 mM KCl, 10 mM $Na_2HPO_4$, 1.8 mM $KH_2PO_4$)—vortex, centrifuge, discard supernatant. Finally, the pellet was resuspended in PBS and stored at 4 °C for further analysis.

**Flow cytometry analysis**. Samples used for flow cytometry analysis were thawed on ice and appropriately diluted in PBS—commonly 1/500, to achieve optimal cell density for flow. Samples were stained for DNA using either SYBR Green I Nucleic acid gel stain Ex504/Em523 (10,000× concentrate in DMSO—subsequently diluted to 1:1000) (Invitrogen, Thermo Fisher, UK) or SYTO 9 Green Fluorescent Nucleic acid stain Ex485/Em498, diluted 1:1000 (Invitrogen, Thermo Fisher, UK). Sample solutions were flowed on a Beckman Coulter Moflo Astrios Eq, typically collecting 50–100,000 events per sample (minimum of 10,000 was used when analyzing fermentations demonstrating minimal growth) and at a constant flow rate (slow). All data was processed using the Kaluza analysis software v.2.1. Firstly, for each bacterial species a "negative" sample (24 h growth on unlabelled substrate) was flowed in order to establish a gating protocol as to only process events/cells which were positive for SYBR/SYTO DNA staining (488-FSC2 Height predominantly used). Samples were then deemed to be "fluorescently labelled" or not by a positive shift in the correlation of 488-FSC2 height vs. 355-448/59 height (fluorescence intensity threshold).

**HPLC supernatant analysis**. For the analysis of 2-AB-labelled standards as well as recovered fermentation supernatants hydrophilic interaction chromatography (HILIC), was applied using an Agilent 1290 Infinity (Agilent Technologies, Santa Clara, CA, USA) UHPLC system. The methodology was based on that developed by Westereng et al. 2020[72], with the following modifications. The system was connected in parallel (1:10 split) to an Agilent 1260 fluorescence detector—Ex 330 nm, Em 420 nm (Agilent Technologies, Santa Clara, CA, USA) and a Velos pro LTQ linear ion trap (Thermo Scientific, San Jose, CA, USA). The HILIC column (bioZen Glycan, 2.6 μm, 2.1 × 100 mm) including a guard column (SecurityGuard ULTRA with bio-Zen Glycan cartridge, 2.1 × 2 mm) was operated at 50 °C, running at 0.3 mL/min, and using 50 mM ammonium formate pH 4.4 (eluent A) and 100% acetonitrile (eluent B). Samples were eluted using the following gradient: initial starting ratio of 90% B and 10% A, gradient to 72% B and 28% A from 0 to 16 min, gradient to 40% B and 60% A from 16 to 20 min, isocratic from 20 to 25 min, gradient to 76% B and 24% A from 25 to 27 min, isocratic from 27 to 30 min. Two μL of the samples were injected.

**HPLC-HILIC-FLD (MS) of substrates**. For HILIC-FLD-MS the instrument was operated in positive mode with an ionization voltage of 3.5 kV, auxiliary and sheath gas settings of 5 and 30, respectively (arbitrary units), and with capillary and source temperatures of 300 and 250 °C, respectively. The scan range was set to $m/z$ 110–2000 and MS/MS analysis was performed with CID fragmentation with helium as the collision gas. All data were recorded with Xcalibur version 2.2.

**MALDI-ToF**. MALDI-ToF analyses were conducted using an Ultraflex MALDI-ToF/ToF instrument (Bruker Daltonics, Germany) equipped with a 337-nm-wavelength nitrogen laser. All measurements were performed in positive mode. Data were collected from 100 laser shots using the lowest energy level necessary to obtain sufficient spectral intensity. The mass spectrometer was calibrated with a mixture of manno-oligosaccharides or xylo-oligosaccharides (DP, 2–6) obtained from Megazyme. For sample preparation, 1 μL of sample solution was mixed with 2 μL of matrix (9 mg/mL 2,5-dihydroxybenzoic acid (DHB)–30% acetonitrile v/v), directly applied to a MTP 384 target plate (Bruker Daltonics, Germany), and dried under a stream of warm air.

**Statistics and reproducibility**. All experiments were carried out in biological duplicates.

**Reporting summary**. Further information on research design is available in the Nature Research Reporting Summary linked to this article.

## Data availability

All data supporting the findings of this study are available within the article and supplemental material. Source data underlying figures are presented in Supplementary Data 1.

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

## Acknowledgements

We are grateful for support from The Research Council of Norway (project numbers; 244259, 295501, 270038, 295910 and 319049).

## Author contributions

S.L and L.L. performed flow cytometry and confocal microscopy. P.G. conducted parts of the growth experiments. S.L. carried out all other analysis. S.L. and B.W. conceived the study and supervised the research. The manuscript was primarily written by S.L. with contributions from B.W., S.L.L.R., and L.L. Figures were prepared by S.L.

## Competing interests

The authors declare no competing interests.
