## [Peer Review File · Communications Biology]

Reviewers' comments:

Reviewer #1 (Remarks to the Author):

This manuscript describes a useful technology that expands that micro/glycobiologist tool kit of labeled glycan strategies for probing utilization by bacterial communities. The work here describes how the reducing end of different poly/oligosaccharide can be labeled with 2-aminobenzamide (2-AB), a small fluorescent probe. This strategy not only provides a much smaller fluorescent label than other previously reported methods but also eliminates the random labeling that occurs with other chemistries. The authors use a rigorous array of methods to demonstrate their methods including HPLC-HILIC-FLD, mass spec, microscopy and FACS to demonstrate the utility of this method. Overall this is a well conducted study. I have a few suggestions for the authors in order to improve the clarity of the manuscript and perhaps glean more biological context out of the experiments that were performed here.

1. General comment: I realize that goal of this report is to demonstrate the utility of the 2-AB probe for analyzing glycan uptake by bacteria in general and not to comment on the glycan uptake capabilities of the specific bacteria used here. But that said, I do think the authors need to provide some text explaining their rationale for using *B. cellulosilyticus* and the other *Bacteroides* strains, including their ability to utilize mannan and xylan (and reference the studies describing this.) I think this is also important for evaluating the specificity of uptake, especially with regards to Figure 8/Supplemental Figure 5. I also think it's appropriate to have a little more discussion about the biological results from this study, to frame how growth/uptake of the labeled glycan correlate with what is known about how these bacteria utilize mannan or xylan.

2. Another general comment that I have is that I don't see any actual bacterial growth data here. It looks like the proportion of labeled cells increases in different experiments but without knowing if the actual cell mass used or if culture turbidity is changing then this doesn't in and of itself demonstrate growth.

3. The substrates AcGGM and AcAGX are spelled out and explained in the methods, but I think it's important to spell these out and very briefly describe why they were chosen in the beginning part of the results. This includes how the GH-substrates were generated.

4. Lines 165-167 state that the methodology for labeling extended to a wide range of other glycans – what glycans were tested?

5. Lines 184-189: Why was condition 2 most successful? It is interesting that going from glucose grown to all labeled substrate did not permit (much? any?) cell growth. None of the data for these trials is shown. I would ideally like to see what the growth curves are for some of these different conditions. Which bacterial cultures were used (single species?) and which labeled substrates in these trials? The information here is very vague. I assumed GGM and xylooligos? Did the cultures perform the same way for both oligos?

6. Lines 193 – 196: To clarify, the authors saw that there was uptake of the label upon exposure to cells? What is meant by carbohydrate degradation here – is the labeled compound breaking down in the culture?

7. Lines 205 – 207: Does “numerous experiments” refer to replicates of growths or experiments performed with different combinations of bacteria/substrates? Also, the authors describe that the label is taken up very quickly... but cells are not growing on the labeled glycan (ie strategy 1 in Figure 2 was suboptimal)? I guess I am (still) a little confused why glucose + labeled glycan must be added as mentioned above. Do you think this is specific to *Bacteroides*?

8. Lines 212- 214 “The substrate used for comparison was also subjected to the same conditions applied in fermentations (Figure 4)...” Is this data shown? I thought the 24 hour supernatant line in these panels was 24 hours after bacterial growth –as in the spent supernatant?

9. Lines 219-220 referring to single strain or co-culture – is there a specific experiment being

referred to here? The lack of description of the bacterial strain or glycan being used makes the text seem vague.

10. Regarding the use of SYBR and SYTO 9 - is there a reference for the previous use of these dyes for bacterial DNA or a control from this work demonstrating staining of DNA? The same applies to the use of the membrane stains.

11. Lines 246 – 247: The text states that cells labeled with either mannan or xylan were identifiable but only mannan labeled cells shown.

12. Figure 8: I feel the difference in label accumulation here should be accounted for, along with references and a statement that these species are known to utilize these oligosaccharides. On the last note, the incorporation of AGX especially is surprising for Bc and Bt as I don't think the type strains are known to degrade xylan? Or perhaps these strains do? Its not specifically stated. Could this be non-specific labeling, as in labeling independent of uptake for cell growth? Some demonstration of growth/utilization might be appropriate or necessary here to make this point. Between Fig 8 and supplemental fig 5 I think there needs to be more biological explanation for these results. With the way the co-culture data are presented, its unknown which cells in the mix are labeled and/or who grew over time.

13. Lines 263 – 265: Goes a bit with the point above. I feel like more should be said here...what are the biological conclusions?

14. Lines 371 – 373: This is the first place in the text in which the species used are described. This information should be presented sooner including why these were studied.

Minor

There is some awkward phrasing in the following manuscript lines:

21 (abstract)

38 (introduction)

111

115 – HILIC should be spelled out the first time I think.

332/342 – should be were instead of was

346 – oil bath or water bath?

PBS all caps

Reviewer #2 (Remarks to the Author):

In this manuscript, Leivers and coworkers deployed a well-established reductive amination approach to deliver a fluorophore to complex wood-derived glycans and then monitor their uptake by gut bacteria grown in monoculture or co-culture. The advance here is a proof-of-principle development of a multi-modality platform for monitoring glycan uptake by bacteria. The authors systematically demonstrate the synthesis of the glycan probes using established methods and the ability to monitor glycan-probe uptake by bacteria via analytical approaches (chromatography/LCMS) coupled to monitoring of cultured cells (flow cytometry and microscopy). The uptake of glycans by a handful of bacterial species was explored. The method presented has utility for monitoring glycan uptake in more complex microbial communities.

Major critique that must be addressed prior to publication:

The results presented showcase the technology but fall short of yielding novel insights into the underlying biology. The work would be considerably strengthened by exploring the physiological impact of glycan utilization on the bacterial species and co-cultures explored. For example, how is growth impacted? Viability? Does selective glycan utilization confer a competitive advantage to some bacterial species? What is the relative fitness of bacteria within mixed co-cultures

supplemented with the different glycans? Addressing these questions is especially important for determining the physiological importance of glycan utilization. Without exploring the fitness effects of glycan utilization, the work is considerably less impactful than it would be by making this link explicit. This work may be suitable for publication in *Communications Biology* if this major critique is addressed experimentally, along with the more minor critiques raised below. In its present form, the work lacks the urgency required for publication in *Communications Biology*.

Additional critiques that should be addressed prior to publication:

1. The labeled glycans are not sufficient energy sources on their own and can only be used when cells are cultured with glucose. However, no data is provided to support this observation/statement (Figure 2). The authors should provide evidence to support their claim.
2. It is unclear what the labeled glycans are being used for. If they are catabolized as energy sources, why is growth with glucose required? Could they be used as raw materials for building up higher-order glycans?
3. The claim within the abstract that the authors' approach allows comprehensive tracking of metabolism of labeled glycans is overstated. The fate of the glycans remains unclear.
4. What happens when bacteria are treated with the 2-AB label absent conjugation to a glycan? This important control should be added alongside treatment with the 2-AB-labeled glycans.
5. The potential exists for the reductive amination-based label to impact uptake of glycans and their subsequent utilization. The authors should address this possibility.
6. The main text uses acronyms for complex glycans (e.g. GH-10-AGX, GH26-AcGGM) but does not directly define these acronyms nor the structures they refer to. The acronyms should be defined in the main text (not just in the experimental section) to aid readability and the structures of the labelled glycan structures should be explicitly included in the supplemental information.
7. The authors assume that the labelled glycans are being taken up and catabolized rather than binding to surface receptors. The flow cytometry and microscopy data cannot distinguish between these possibilities. What additional evidence do the authors have to support their claim? Optimally, microscopy data would be shown in higher resolution/magnification to allow determination of whether the labelled substrates make it inside cells or are instead bound on the cell surface.
8. Pertaining to the results in Figure 8: It would be considerably more interesting if the authors differentiated which cells in the mixed populations uptake the labelled sugars and become fluorescent. What happens if individual populations are tracked within the mixed population? And how does uptake track with relative fitness?

To the Reviewers

Thank you very much for the insightful comments. Although requiring a large amount of additional work, we completed a comprehensive set of growth experiments (consuming several grams of substrate), which clearly improves the study, and demonstrates some of the potential uses of these labelled glycans. We feel these growth experiments both clarify and address several of the reviewer's comments (Reviewer #1 Comments 1,2,5,8,12,13. Reviewer #2 Comments 1,2,4.). Unfortunately, it seems that the line numbers are slightly out of sync from my original uploaded Word copy to the pdf version you received. All line numbers (and subsequent new figures – new Figure 1 added therefore all previous figures have shifted in number) referred to by the authors apply to the revised manuscript (with revisions clearly highlighted).

Reviewers' comments:

Reviewer #1 (Remarks to the Author):

This manuscript describes a useful technology that expands that micro/glycobiologist tool kit of labeled glycan strategies for probing utilization by bacterial communities. The work here describes how the reducing end of different poly/oligosaccharide can be labeled with 2-aminobenzamide (2-AB), a small fluorescent probe. This strategy not only provides a much smaller fluorescent label than other previously reported methods but also eliminates the random labeling that occurs with other chemistries. The authors use a rigorous array of methods to demonstrate their methods including HPLC-HILIC-FLD, mass spec, microscopy and FACS to demonstrate the utility of this method. Overall this is a well conducted study. I have a few suggestions for the authors in order to improve the clarity of the manuscript and perhaps glean more biological context out of the experiments that were performed here.

1. General comment: I realize that goal of this report is to demonstrate the utility of the 2-AB probe for analyzing glycan uptake by bacteria in general and not to comment on the glycan uptake capabilities of the specific bacteria used here. But that said, I do think the authors need to provide some text explaining their rationale for using *B. cellulosilyticus* and the other *Bacteroides* strains, including their ability to utilize mannan and xylan (and reference the studies describing this.) I think this is also important for evaluating the specificity of uptake, especially with regards to Figure 8/Supplemental Figure 5. I also think it's appropriate to have a little more discussion about the biological results from this study, to frame how growth/uptake of the labeled glycan correlate with what is known about how these bacteria utilize mannan or xylan.

As stated by the reviewer, the goal of this report is to demonstrate that the 2-AB probe can be successfully used for labelling carbohydrates (both polymeric glycans and oligosaccharides) and determine specific bacterial glycan uptake. The paper is a proof-of-concept that shows that bacteria known to possess an apparatus for AcGGM and AGX utilization (*B. ovatus* or *B. cellulosilyticus*) are able to utilize the 2-AB labelled substrates; when in co-culture with a known AcGGM and AGX non-utilizing bacterium (*B. caccae* and *B. thetaiotamicron*), the labelled glycans selectively promoted the growth of the known

mannanolytic or xylanolytic bacterium. Appropriate references to the studies describing the ability, or inability, for the selected bacteria to utilize AcGGM and AGX have now been included in the revised version of the manuscript at lines 198-200.

New text:

The bacteria used were selected based on their known ability to utilize β -mannans and xylans (*B. cellulosityticus* and *B. ovatus*²) or the lack of this capability (*B. theta* and *B. caccae*^{2,66}).

2. Another general comment that I have is that I don't see any actual bacterial growth data here. It looks like the proportion of labeled cells increases in different experiments but without knowing if the actual cell mass used or if culture turbidity is changing then this doesn't in and of itself demonstrate growth.

As per the reviewer's comment, growth data has now been included as part of Figure. 3 in the main manuscript and in a new Supplementary Figure. 6. We feel this data now provides appropriate weight to demonstrate both growth on labelled substrates as well as their unlabelled counterparts.

3. The substrates AcGGM and AcAGX are spelled out and explained in the methods, but I think it's important to spell these out and very briefly describe why they were chosen in the beginning part of the results. This includes how the GH-substrates were generated.

A new Figure. 1 has been added to introduce the substrates. The substrates and their modified counterparts are now comprehensively explained at the beginning of the Results and Discussion under the 'Substrate generation & Characterisation' section (Line Numbers 155-162).

Old text:

In this study we used reducing end coupling by reductive amination (via picoline borane) to successfully conjugate 2-AB molecules to β -mannan- and xylan-derived oligosaccharides from Norway spruce and birch wood (Figure 1 a).

New text:

In this study we generate fluorescently labelled carbohydrates using β -mannan- (AcGGM) and xylan-derived (AcAGX) oligosaccharides from Norway spruce and birch wood, respectively (Figure 1), in reducing end coupling by reductive amination (via picoline borane) to successfully conjugate 2-AB molecules (Figure 2 a). Modified versions of each of these substrates were created by treatment with a β -mannanase (*R. intestinalis* β -mannanase RiGH26) for AcGGM generating the shorter GH26-AcGGM and a combined alkaline and xylanase treatment for AcAGX, generating the shorter, deacetylated GH10-AGX.

4. Lines 165-167 state that the methodology for labeling extended to a wide range of other glycans – what glycans were tested?

This section has now been edited to specifically state glucose (monomer), lactose (disaccharide) and other enzymatically and chemically modified mannans and xylans (oligosaccharides) not applicable for this study. A number of references to 2-AB labelling of carbohydrates are also included in the section 'A New Approach for Microbial Screening – Smaller, Faster, Cheaper' (Line Numbers 167-169).

Old text:

The study also extended to a number of other monomers, oligo- and polysaccharides, observing that this approach is applicable to a wide range of glycans with reducing ends.

New text:

The study also extended to a number of other monomers (glucose), disaccharides (lactose) and oligosaccharides (enzymatically and chemically modified mannans and xylans), observing that this approach is applicable to a wide range of glycans with reducing ends.

5. Lines 184-189: Why was condition 2 most successful? It is interesting that going from glucose grown to all labeled substrate did not permit (much? any?) cell growth. None of the data for these trials is shown. I would ideally like to see what the growth curves are for some of these different conditions. Which bacterial cultures were used (single species?) and which labeled substrates in these trials? The information here is very vague. I assumed GGM and xylooligos? Did the cultures perform the same way for both oligos?

We hope the addition of growth data (Figure 3 b and Supplementary Figure 6) helps to support the decision to choose condition set 2 for the experiments carried out. For further clarification, this set up was selected as it enabled the researchers to show the specific lack of growth of *B. Caccaae* and *B. theta* on the chosen labelled substrates (and unlabelled in the additional growth curves) but still achieve growth overall (due to the glucose present) as to have sufficient cells for flow cytometry experiments, to analyse labelled and unlabelled cell populations.

We have also tried to emphasise this further by adding this following text to the Fermentation & Growth section (Line Numbers 188-196).

Old text:

Condition set **2**, consisting in a mixture of glucose and labelled substrate, was chosen going forwards as it provided the best resolution for flow cytometry measurements, as poor or in some instances, no growth on labelled and/or unlabelled substrates (conditions **1**, **3** and **4**) resulted in a lack of viable cells to analyse.

New text:

Condition set **2**, consisting of a mixture of glucose (on which all bacteria grew effectively) and labelled substrate, was chosen going forwards as it provided the best resolution for flow cytometry measurements, as poor or in some instances, no growth on labelled and/or unlabelled substrates (conditions **1**, **3** and **4** (for *B. caccae* and *B. theta*)) resulted in a lack of

viable cells to analyse.

6. Lines 193 – 196: To clarify, the authors saw that there was uptake of the label upon exposure to cells? What is meant by carbohydrate degradation here – is the labeled compound breaking down in the culture?

For clarity this has been reworded to carbohydrate catabolism rather than degradation (Line numbers 204-206). There is no evidence of labelled compound breakdown in the culture solution when no bacteria are present.

Old text:

this was due to rapid carbohydrate degradation and label uptake, even within the relatively short time period of exposure during sampling.

New text:

this was due to rapid labelled carbohydrate uptake and catabolism, even within the relatively short time period of exposure during sampling.

7. Lines 205 – 207: Does “numerous experiments” refer to replicates of growths or experiments performed with different combinations of bacteria/substrates? Also, the authors describe that the label is taken up very quickly... but cells are not growing on the labeled glycan (ie strategy 1 in Figure 2 was suboptimal)? I guess I am (still) a little confused why glucose + labeled glycan must be added as mentioned above. Do you think this is specific to Bacteroides?

Numerous experiments simply refers to the general trend observed from cell tracking by flow cytometry and supernatant analysis by HPLC-HILIC-FLD. We hope the rest of this comment, regarding growth strategy is explained sufficiently in the reply to comment 5 above.

8. Lines 212- 214 “The substrate used for comparison was also subjected to the same conditions applied in fermentations (Figure 4)...” Is this data shown? I thought the 24 hour supernatant line in these panels was 24 hours after bacterial growth –as in the spent supernatant?

This section of the text along with description of what is now Figure 5, has been amended (Line numbers 222-226) to more clearly specify that there were no discernible differences in substrate pre-fermentation and substrate included in culture media (no bacteria present, subjected to the same atmospheric conditions). An example of this is provided in Supplementary Figure 5 c, where no degradation of the substrate is observed (as expected) after up to 72 hours by *B. caccae* + *B. theta*.

Old text:

The substrate used for comparison was also subjected to the same conditions applied in fermentations (Figure 4), thus confirming that any observable degradation occurred as a

direct result of bacterial processing of the carbohydrates and not due natural degradation over time from exposure to the controlled anaerobic environment.

New text:

The substrate used for comparison was also subjected to the same conditions applied in fermentations, with the almost identical chromatogram (compared to T0 chromatogram) observed thus confirming that any observable degradation occurred as a direct result of bacterial processing of the carbohydrates and not due to natural degradation over time from exposure to the controlled anaerobic environment.

9. Lines 219-220 referring to single strain or co-culture – is there a specific experiment being referred to here? The lack of description of the bacterial strain or glycan being used makes the text seem vague.

This text has been made clearer (Line numbers 229-233), with more specific descriptions provided relating to the figures in question (Figure 5., Figure 6 c, Supplementary Figure 4 a-b and 5 b showing single strains and Supplementary Figure 5 c showing co-cultured strains).

Old text:

In either single strain or co-cultured fermentations, the degradation of substrates, observed by the shift of oligosaccharide DP from high to low, was clearly demonstrated through chromatographic and mass spectrometry techniques.

New text:

In either single strain or co-cultured fermentations, the degradation (or lack thereof) of substrates, observed by the shift of oligosaccharide DP from high to low, was clearly demonstrated through chromatographic and mass spectrometry techniques (Single strain - Figure 5, Figure 6 c and Supplementary Figure 4 a-b and 5 b, Co-cultured – Supplementary Figure 5 c).

10. Regarding the use of SYBR and SYTO 9 - is there a reference for the previous use of these dyes for bacterial DNA or a control from this work demonstrating staining of DNA? The same applies to the use of the membrane stains.

Appropriate references have been added to the Confocal Microscopy section (Line numbers 247-249) which concern the use of all stains/dyes used in the analysis.

Old text:

Two alternative dyes were therefore investigated for DNA staining including SYTO 9 and SYBR Green I, along with two potential stains for membrane identification (FM 5-95 and BacLight Red).

New text:

Two alternative dyes were therefore investigated for DNA staining including SYTO 9 and SYBR Green I^{67,68}, along with two potential stains for membrane identification (FM 5-95⁶⁹

and BacLight Red⁷⁰).

11. Lines 246 – 247: The text states that cells labeled with either mannan or xylan were identifiable but only mannan labeled cells shown.

Xylan (AcAGX) has been added as an additional panel in Figure.8 (previously Figure. 7).

12. Figure 8: I feel the difference in label accumulation here should be accounted for, along with references and a statement that these species are known to utilize these oligosaccharides. On the last note, the incorporation of AGX especially is surprising for Bc and Bt as I don't think the type strains are known to degrade xylan? Or perhaps these strains do? Its not specifically stated. Could this be non-specific labeling, as in labeling independent of uptake for cell growth? Some demonstration of growth/utilization might be appropriate or necessary here to make this point. Between Fig 8 and supplemental fig 5 I think there needs to be more biological explanation for these results. With the way the co-culture data are presented, its unknown which cells in the mix are labeled and/or who grew over time.

In a previous study (La Rosa, S. L. *et al.* Wood-Derived Dietary Fibers Promote Beneficial Human Gut Microbiota. *mSphere* 4, (2019), we have shown that *B. cell* is able to utilize both AcGGM and AcAGX while *B. caccae* and *B. theta* are not. In the mixture *B. ovatus* + *B. caccae* and *B. ovatus* + *B. theta*, or *B. cell* + others it is likely that only *B. ovatus* or *B. cell* (Figure 9 & Supplementary Figure 5) are growing (as also confirmed by our additional growth data). We also feel this data rules out non-specific labelling, as suggested above.

We have provided a number of viable explanations for the difference in label incorporation from 24 to 72 hours shown in Supplementary Figure 5 (see below for additional text). We also agree that the small amount of label incorporation observed (below 10%) in *B. theta* + *B. caccae* mixed culture at 72 hours is a little surprising considering their inability to utilize the substrate, however this may be explained along the same lines as co-growth, by slow/inefficient uptake of the few oligosaccharides that are present in the AcAGX. This is also supported by the slightly higher growth (still relatively low) observed in the GH10-AGX growth curves compared to other substrates (Supplementary Figure 6). Therefore, this small amount of accumulation of label may be explained by the small amount of short xylooligosaccharides also present in the AcAGX. Considering the chromatograms in panel C, which show that the substrate profile does not change over time, demonstrating that the two strains are not utilizing the labelled AcAGX to any significant degree.

Old text:

There are numerous hypotheses for this, two of which are i) adaptation of the bacteria over time frame of the experiment; ii) cross-feeding of the different strains – *B. cellulosilyticus* producing more readily accessible carbohydrates for other organisms, which has been shown recently to be a viable possibility¹.

New text:

There are several possible explanations for this, three of which are i) *B. cellulosilyticus* in combination with other strains causes a delay to label uptake; ii) adaptation of the bacteria to utilize the substrate over the time frame of the experiment, hence an extended lag phase due to suboptimal growth conditions; iii) cross-feeding of the different strains – *B. cellulosilyticus* producing more readily accessible carbohydrates for other organisms, which has been shown recently to be a viable possibility¹. Whilst we also recognise that the small amount of growth after 72 hours in the *B. caccae* + *B. theta* is unexpected, it may be due to some shorter oligosaccharides present that can be taken up via other uptake channels other than the specific xylan transporter.

13. Lines 263 – 265: Goes a bit with the point above. I feel like more should be said here...what are the biological conclusions?

Hopefully we have sufficiently suggested an explanation for this above.

14. Lines 371 – 373: This is the first place in the text in which the species used are described. This information should be presented sooner including why these were studied.

For clarification we have now introduced the specific strains used (along with reasoning for using them) in this study at a much earlier stage in the manuscript (Line numbers 198-200), towards the start of the Results & Discussion in the T0 sampling section. This new text has previously been referenced in the reply to Comment 1.

Minor

There is some awkward phrasing in the following manuscript lines:

21 (abstract) – Phrasing altered (Line number 21).

Old text:

Studying specific glycan uptake and metabolism

New text:

The study of specific glycan uptake and metabolism

38 (introduction) – We have looked through the text and could not find any ambiguities here and are thus uncertain about what the reviewer is referring to. No changes made.

111 – As above.

115 – HILIC should be spelled out the first time I think. – Amended in manuscript.

332/342 – should be were instead of was – Amended in manuscript.

346 – oil bath or water bath? – Amended in manuscript.

PBS all caps – Amended in manuscript.

Reviewer #2 (Remarks to the Author):

In this manuscript, Leivers and coworkers deployed a well-established reductive amination

approach to deliver a fluorophore to complex wood-derived glycans and then monitor their uptake by gut bacteria grown in monoculture or co-culture. The advance here is a proof-of-principle development of a multi-modality platform for monitoring glycan uptake by bacteria. The authors systematically demonstrate the synthesis of the glycan probes using established methods and the ability to monitor glycan-probe uptake by bacteria via analytical approaches (chromatography/LCMS) coupled to monitoring of cultured cells (flow cytometry and microscopy). The uptake of glycans by a handful of bacterial species was explored. The method presented has utility for monitoring glycan uptake in more complex microbial communities.

Major critique that must be addressed prior to publication:

The results presented showcase the technology but fall short of yielding novel insights into the underlying biology. The work would be considerably strengthened by exploring the physiological impact of glycan utilization on the bacterial species and co-cultures explored. For example, how is growth impacted? Viability? Does selective glycan utilization confer a competitive advantage to some bacterial species? What is the relative fitness of bacteria within mixed co-cultures supplemented with the different glycans? Addressing these questions is especially important for determining the physiological importance of glycan utilization. Without exploring the fitness effects of glycan utilization, the work is considerably less impactful than it would be by making this link explicit. This work may be suitable for publication in *Communications Biology* if this major critique is addressed experimentally, along with the more minor critiques raised below. In its present form, the work lacks the urgency required for publication in *Communications Biology*.

Thank you for this comment. We have conducted a number of supporting growth experiments which we hope help answer the aforementioned concerns/queries. We acknowledge that these new growth experiments underscore: that Bacteroidetes with a degradation machinery for mannan and xylan (grow well) on the substrate (very similar to non-labelled substrate). This implies that these substrates could be successfully used as a labelling technique applicable for growth experiments also in more complex consortia. Due to the high sensitivity of the fluorophore, you may reduce the proportion of fluorescent labelled substrate (to reduce cost) to minimize potential impact on. Furthermore, the use of fluorescent labels in combination with cell sorting can foreseeably considerably reduce the workload/as well as improving resolution on more expensive and more time consuming omics techniques – by sorting out only the microbes that take up the labelled glycan, hence the screening terminology used to describe this work.

Additional critiques that should be addressed prior to publication:

1. The labeled glycans are not sufficient energy sources on their own and can only be used when cells are cultured with glucose. However, no data is provided to support this observation/statement (Figure 2). The authors should provide evidence to support their claim.

We apologize this ambiguity in the text and have specified in the manuscript accordingly. We hope the reply from comment 5 of reviewer 1 sufficiently also answers this query.

2. It is unclear what the labeled glycans are being used for. If they are catabolized as energy sources, why is growth with glucose required? Could they be used as raw materials for building up higher-order glycans?

Again, we apologize for this ambiguity in the text. We hope this comment is addressed above (also in the reply from comment 5 of reviewer 1) in regard to making clearer the need for the for growth of all bacteria (using glucose supplemented fermentations).

3. The claim within the abstract that the authors' approach allows comprehensive tracking of metabolism of labeled glycans is overstated. The fate of the glycans remains unclear.

We understand the reviewer's issue with this, as such the phrasing 'comprehensively track' has been changed to - better understand (Line number 29).

Old text:

it is possible to comprehensively track

New text:

it is possible to better understand

4. What happens when bacteria are treated with the 2-AB label absent conjugation to a glycan? This important control should be added alongside treatment with the 2-AB-labeled glycans.

Thank you for pointing out this. We hope this question has been answered by the additional growth data, particularly in the new panel added to Figure 3.

5. The potential exists for the reductive amination-based label to impact uptake of glycans and their subsequent utilization. The authors should address this possibility.

These concerns have been addressed as a possibility and elaborated on in the section titled 'A Technique for Screening Microbial Communities/Substrate Specificity' (Line numbers 283-286).

New text:

Whilst we have not experienced any impact on uptake, imparted by the addition of the 2-AB label (where most growth curves for bacteria with labelled and unlabelled substrate show relatively comparable optical densities (OD), Supplementary Figure 6, the potential for this to cause an issue will require further analysis.

6. The main text uses acronyms for complex glycans (e.g. GH-10-AGX, GH26-AcGGM) but does not directly define these acronyms nor the structures they refer to. The acronyms should be defined in the main text (not just in the experimental section) to aid readability and the structures of the labelled glycan structures should be explicitly included in the supplemental information.

We hope the reply from comment 3 of reviewer 1 sufficiently also answers this query.

7. The authors assume that the labelled glycans are being taken up and catabolized rather than binding to surface receptors. The flow cytometry and microscopy data cannot distinguish between these possibilities. What additional evidence do the authors have to support their claim? Optimally, microscopy data would be shown in higher resolution/magnification to allow determination of whether the labelled substrates make it inside cells or are instead bound on the cell surface.

Thank you for this comment. Unfortunately, we have limited access to highly sophisticated microscopy, and it was not possible to obtain such data the reviewer asked for. However, we have conducted additional growth data on labelled substrates as the sole carbon source showing solid growth. Thus, we hope that the newly provided growth data (Figure 3 b and Supplementary Figure 6), which shows growth on the labelled substrates without glucose present along with the original data which shows the change in the structure of the labelled substrate over time (bacterial catabolism) is sufficient evidence to support our claim.

8. Pertaining to the results in Figure 8: It would be considerably more interesting if the authors differentiated which cells in the mixed populations uptake the labelled sugars and become fluorescent. What happens if individual populations are tracked within the mixed population? And how does uptake track with relative fitness?

Whilst we clearly agree with this comment and see a large potential for this methodology, we feel this work is clearly out of scope of this study, but that this builds the foundation for future studies on more complex mixed cultures.

REVIEWERS' COMMENTS:

Reviewer #2 (Remarks to the Author):

I have carefully reviewed the authors' rebuttal letter, changes to the manuscript, and expanded supplementary data. Based on this assessment, I conclude that they rigorously and thoroughly addressed all reviewer concerns and critiques. Their manuscript is significantly strengthened as a result and is appropriate for publication in *Communications Biology* in its current form. The described microbial glycobiochemistry toolkit will be of much value to the field.